# A national survey of videolaryngoscopes and alternative intubation devices in Hungary

Bálint Nagy[1,2,3]*, Szilárd Rendeki[1,2,3]

**1** Department of Anesthesiology and Intensive Therapy, Medical School, University of Pécs, Pécs, Hungary, **2** Department of Operational Medicine, Medical School, University of Pécs, Pécs, Hungary, **3** Medical Skills Lab, Medical School, University of Pécs, Pécs, Hungary

\* balintjanosnagy@yahoo.com

## Abstract

### Introduction

Videolaryngoscopy (VL) as a new airway management technique has evolved in recent decades, and a large number of videolaryngoscopes are now available on the market. Most recent major guidelines already recommend the immediate availability and use of VL in difficult airway management scenarios. However, national data on the availability of VL, introduction into practice and patterns of use are rarely published. Therefore, the current study aimed to provide data on VL in Hungary.

### Materials and methods

An electronic survey was designed and popularized with the help of the Hungarian Society of Anaesthesiology and Intensive Therapy to explore the availability, use, and practice of and attitudes toward VL among Hungarian anesthesiologists. The survey was conducted between 01.01.2018 and 31.12.2018.

### Results

In total, 324 duly completed forms were returned and analyzed. Responders were mainly males (58%), specialists (80%) and those involved mainly in anesthesia practice (68%) in the public sector. Two hundred and ten (65%) responders had access to various videolaryngoscopes and were mainly from surgery, intensive care and traumatology units. No responders reported the availability of eight videolaryngoscopes out of the eighteen listed devices, and 32% of the responders had never used any videolaryngoscope in clinical settings. The most commonly available devices were KingVision, MacGrath Mac and Airtraq. Most of the responders reported using videolaryngoscopes mainly for difficult airway management and reported using a fiberscope as the first alternative device. Popular methods for selecting videolaryngoscopes included the following: short clinical trial (n = 67/324), decision of the departmental lead (n = 65/324) and price (n = 54/324). The majority of responders had some training prior to clinical application, but training was mainly voluntary. Overall, 98% of the responders considered videolaryngoscopes beneficial.

**Data Availability Statement:** The datasets used and/or analyzed during the current study are available as supporting information of the manuscript.

**Funding:** The study was supported by a "Postdoc" scholarship, courtesy of the Medical School, University of Pécs, Hungary. This study was also supported by EFOP-3.6.3-VEKOP-16-2017-00009 and EFOP-3.6.1-16-2016-00004 through the Medical School, University of Pécs, Hungary and by the Biomedical Engineering Project (TUDFO/51757-1/2019-ITM) at University of Pécs, Hungary. The funders had no role in study design, data collection and analysis, decision to publish, or preparation of the manuscript.

**Competing interests:** The authors have declared that no competing interests exist.

## Conclusions

Approximately two-thirds of Hungarian anesthesiologists have immediate access to video-laryngoscopes, which are used mainly for difficult airway management. The overall attitude towards VL is positive, and many videolaryngoscopes are known and have been used by Hungarian anesthesiologists. However, only a few devices on the market are used commonly. Based on the results, further improvement might be recommended regarding VL training and availability.

## Introduction

Direct laryngoscopy remains the gold standard for endotracheal intubation. However, video-laryngoscopy (VL) as an expanding technology has evolved and become increasingly popular in the last 10 years[1]. The popularity of VL increased due to promising results in terms of a superior laryngeal view, fewer failed intubations and higher success rates than direct laryngoscopy even when used as a rescue technique[2–4]. The use of VL has been recommended for both difficult and routine airway management in many different settings[5–7]. Most recent major European and American guidelines already recommend the use of VL as a part of difficult airway management algorithms[8–10]. Furthermore, according to the latest Difficult Airway Society (DAS) Difficult Intubation Guidelines, it is recommended that VL be immediately available wherever intubation is performed[8]. Although patients may benefit from the availability of VL, the real clinical availability of this technology might be variable even in developed countries. A recent national survey conducted by Cook and Kelly in the United Kingdom (UK) showed that the availability of VL might range between 14–91% depending on clinical areas[11]. Since data on the availability of VL are rarely published, our primary objective was to explore national data on the availability of VL, introduction into practice and patterns of use in Hungary to gain data on the proportion of anesthesiologist using VL, the most used VL, and the time needed to have VL readily available in clinical settings.

## Materials and methods

Prior to this study, permission was first obtained from the Ethics Committees of the Medical Research Council of Hungary (National Healthcare Services Center, Ministry of Human Capacities of Hungary, 28230-2//2018/EKU). Questions relevant to the availability, use and introduction of VL are shown respectively in English and in Hungarian as supporting information in S1 and S2 Appendixes. The survey was designed as a Google form by the author and piloted with the help of the anesthesiologists (n = 67) from the Department of Anesthesiology and Intensive Therapy, Medical School, University of Pécs, Hungary. The survey was conducted between 01.01.2018 and 31.12.2018. We aimed to reach all the 1567 anesthesiologists of Hungary. A link was distributed electronically with the help of the Hungarian Society of Anaesthesiology and Intensive Therapy, and the participants were requested to complete the survey online. Informed consent regarding participation and publishing was obtained from the participants through a question of the questionnaire. The survey asked for single and individual responses from all the anonymous responders. The study presumed that the connection between the patient and the device used for airway management is the anesthesiologist. Therefore, in the current study, the anesthesiologists were asked to answer as individuals in contrast to similar previous studies in which departments or hospitals responded. Anesthesiology and intensive therapy is a combined, five years long training program in Hungary, thus all the anesthesiologists are intensive care physicians as well. In this study, we collected answers from

anesthesiologists and anesthesiology trainees only, even though we aware of the fact, that other physicians like emergency and internal medicine doctors might occasionally use VL for advanced airway management. Although, still anesthesiologists are responsible for advanced airway management is Hungary in the vast majority of cases.

The following devices were included in this survey:

- Airtraq (Prodol Meditec, Guecho, Spain)

- AP Venner (Venner Medical GmbH, Dänischenhagen, Germany)

- Bonfils (Karl Storz, Slough, UK)

- Bullard (Circon, ACMI, Stamford, CT, USA)

- C-MAC (Karl Storz, Slough, UK)

- C-MAC D-blade (Karl Storz, Slough, UK)

- Coopdech (Daiken Medical, Osaka, Japan)

- C-Trach (previously, Laryngeal mask company, Henley-on-Thames, UK)

- GlideScope (Verathon UK, Amersham, UK)

- King Vision VL (Ambu, St Ives, UK)

- Levitan FPS (Clarus Medical, Minneapolis, MN, USA)

- McGrath 5 (Aircraft Medical, Edinburgh, UK)

- McGrath Mac (Aircraft Medical, Edinburgh, UK)

- Pentax AWS (Pentax, Tokyo, Japan)

- Shikani intubating stylet (Clarus Medical, Minneapolis, MN, USA)

- Upsherscope (Mercury Medical, Clearwater, FL, USA)

- Vividtrac (Vivid Medical, Palo Alto, USA)

- Wuscope (Pentax Precision instruments, Orangeburg, NY, USA)

Other answer options also included "none of the above" or "other VL device". We would like to emphasize here that not all of the aforementioned devices are classic videolaryngoscopes. Bonfils, Levithan and Shikani are optical/digital stylets, Upsherscope, Bullard and WuScope are modified classic laryngoscopes, while C-trach is also a different kind of intubation device. However, to avoid confusion, we prefer to refer these devices also as VL's throughout this study similarly to a recent major evaluation of Cook and Kelly[11].

### Statistical analysis

Data were first exported as a Microsoft Excel 2013 (Microsoft Corporation, Redmond, WA, USA) spreadsheet, and then the Statistical Package for the Social Sciences (SPSS) Statistics software, version 25.0 (IBM Corporation, Armonk, NY, USA), was used for further analysis. Data are presented as the mean and standard deviation (SD) or as raw numbers (n) and percentages (%).

## Results

In total, 324 completed forms were returned without duplicates (S1 Table). Response rate was 21%. The mean age of responders was 43 years, and males were slightly overrepresented

(58%). The majority of responders (80%) were specialists, and responders were mainly involved in anesthesia (68%). Different levels of patient care were similarly represented, with the exception of the private sector. Approximately 78% of responders reported being involved in the education of trainees at least once per month. The detailed characteristics of the responders are shown in Table 1.

## Availability of videolaryngoscopy

Two hundred and ten (65%) responders provided positive information on the availability of any type of VL at at least one anesthesia workstation at their main workplaces. Nineteen anesthesiologists (6%) reported having definite access to VL but were unable to name the exact location (clinical area) of the device. Regarding immediate availability, the most well supplied clinical areas were surgery (n = 115, 36%), the intensive care unit (n = 98, 30%) and traumatology (n = 90, 28%) (Fig 1.). The poorest availabilities were reported in the pediatric (n = 21, 7%), emergency (n = 23, 7%) and ear-nose-throat (n = 34, 11%) units. The overall average immediate availability rate was 18%. When the time window for availability was increased to within ten minutes, the overall average availability rate increased with 5% to 23%. By increasing the time window, the best supplied clinical areas remained the same, but the order changed: intensive care unit (n = 143, 44%), surgery (n = 116, 36%) and traumatology (n = 98, 30%). No responders reported availability of the following videolaryngoscopes at all: the AP Venner, Bullard, Coopdech, C-Trach, Levitan, Shikani, Upsherscope and Wuscope.

**Table 1. Characteristics of responders (n = 324).**

| | | |
|---|---|---|
| **Age in years, mean (SD)** | | 43 (11) |
| **Gender, n (%)** | Male | 188 (58) |
| | Female | 136 (42) |
| **Professional experience, n (%)** | Trainee with < 2 years | 15 (5) |
| | Trainee with 2–5 years | 50 (15) |
| | Specialist with < 10 years | 49 (15) |
| | Specialist with 10–20 years | 95 (29) |
| | Specialist with >20 years | 115 (36) |
| **Professional activity, n (%)** | Anesthesia | 219 (68) |
| | Intensive therapy | 97 (30) |
| | Other (patient related) | 4 (1) |
| | Education | 2 (1) |
| | Administration | 0 (0) |
| | Other (non-patient-related) | 2 (1) |
| **Place of work, n (%)** | City/community hospital | 112 (35) |
| | County hospital | 88 (27) |
| | University hospital | 102 (32) |
| | Private hospital | 9 (3) |
| | Other | 13 (4) |
| **Teaching activity, n (%)** | Once per week | 92 (28) |
| | Once per month | 64 (20) |
| | Less than once per month | 96 (30) |
| | No involvement | 72 (22) |

Data are reported as the mean and standard deviation (SD) or as raw numbers (n) and percentages (%).

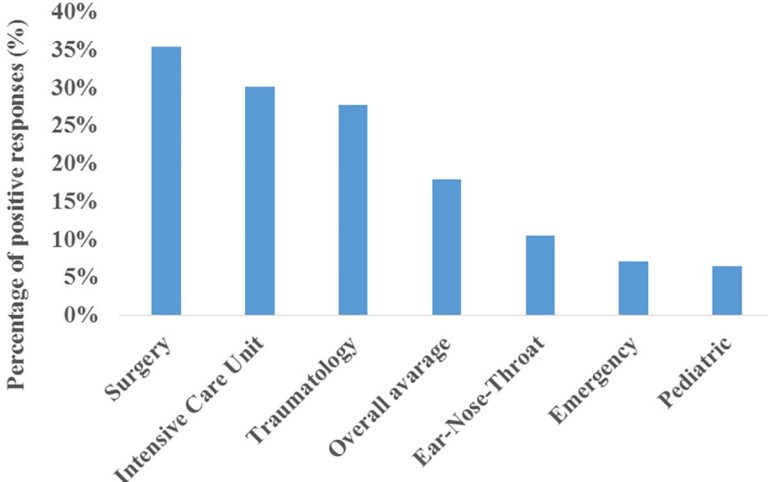

**Fig 1. Immediate availability regarding clinical areas.** The most and the least supplied clinical areas with immediate availability of VL in Hungary according to this survey and based on positive answers given to the following question: "At which workstation at your workplace do you have a videolaryngoscope immediately/readily available? (Option for multiple answers!)".

## Popularity of different videolaryngoscopes

Forty-five (14%) responders declared that they were not familiar with any of the devices included in this survey. The ten most well-known devices are shown in Fig 2. Regarding the real clinical availability of certain videolaryngoscopes the survey showed that only three devices reached at least a 5% positive response. The KingVision was the most available video-laryngoscope in clinical practice at 24% (n = 79), while the McGrath Mac (n = 36, 11%) and Airtraq (n = 28, 9%) were also the part of the top three most common videolaryngoscopes (Fig 3.). Fifty-three (16%) responses reported the following: "A videolaryngoscope is available, but I am not sure about the brand."

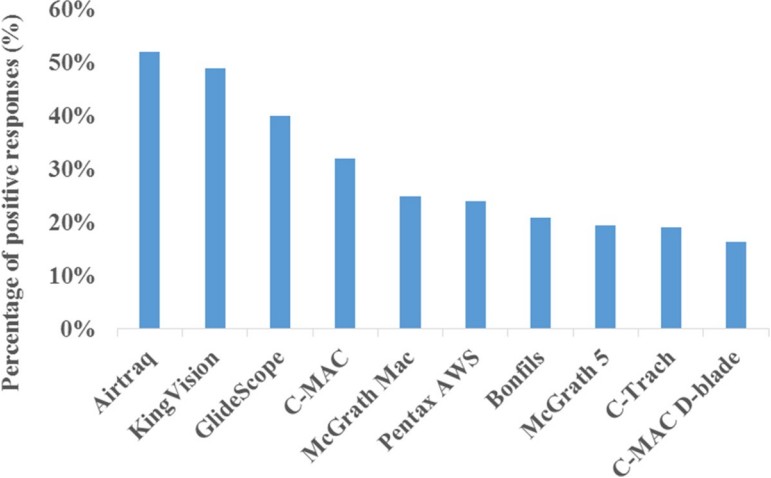

**Fig 2. The most well-known devices in Hungary.** The ten most well-known videolaryngoscopes in Hungary according to this survey and based on positive answers given to the following question: "Have you ever heard about any of the following devices? (Option for multiple answers!)".

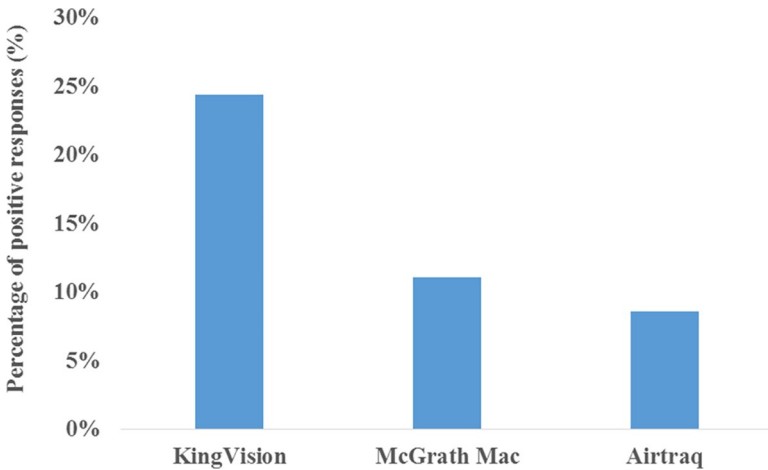

**Fig 3. The most available devices in Hungary.** The three most available videolaryngoscopes in Hungary according to this survey and based on positive answers given to the following question: "Which of the following devices are available at your workplace? (Option for multiple answers!)".

### Patterns of use

One hundred and four responders (32%) said that they had never ever used any videolaryngoscope in clinical settings. Only 39% (n = 126) confirmed that they used VL at least once per month. The KingVision, Airtraq and MacGrath Mac were the top videolaryngoscopes used at least once in patient care by the responding Hungarian anesthesiologists. The following devices were not reported to be used in clinical settings: the Coopdech, Shikani, Upsherscope and Wuscope. The vast majority of users prefer to use VL in "predicted" (n = 151, 47%) and "unexpected" (n = 119, 37%) difficult airway scenarios. The most common indications for VL were the following: "difficulties visualizing the vocal cords appropriately" (n = 303, 94%), "suspected or definitive cervical spine injury" (n = 252, 78%) and "difficulties in endotracheal tube placement even though the vocal cords are fully visible" (n = 153, 47%). Only 11% (n = 37) used VL for "routine" airway management, and 28% (n = 90) used VL for teaching purposes. Fibroscopy was the most popular clinical alternative to VL (n = 281, 87%), while direct laryngoscopy (n = 142, 44%) was the second most common, followed by the use of a laryngeal mask (n = 115, 36%).

### Choice of videolaryngoscopes, education and overall experience

The most common known methods for selecting a videolaryngoscope were the following: short clinical trial (n = 67), decision of the departmental lead (n = 65) and price (n = 54). The majority of users (n = 218, 67%) received some type of training regarding VL. However, training was reported to be mainly on voluntary (n = 187) and rarely compulsory (n = 31) basis. Forty-one (13%) anesthesiologists used VL without any prior training. The overall experience was positive. Excluding those who reported a lack of experience (n = 74, 23%), 98% (n = 246) considered VL beneficial. However, the vast majority of the latest group (n = 210, 65%) found VL useful only under "special circumstances".

### Discussion

Our primary objective was to provide insight into the availability of VL, introduction into practice and patterns of use in Hungary. To our knowledge, no similar evaluation has been

performed regarding VL in Hungary. Therefore, our results might be helpful in many aspects, although our study has several limitations. First, in the current study, the anesthesiologists were asked to answer as individuals in contrast to similar previous studies in which departments or hospitals responded[11,12]. Individual answers were also utilized and found to be interesting in a previous report[13]. Of note, in Gill's study, there was a marked difference between hospital and individual responses regarding VL[13]. The second major limitation might be related to the low response rate. According to the latest data issued by the National Healthcare Services Center of Hungary, 1567 medical doctors have a license to practice as an anesthesiologist in Hungary. Even though fewer doctors might actually be involved in daily anesthesia care, the response rate in this study was still low and estimated to be 20–25%. In a recent similar study by Gill et al., the response rate was 23% for duly completed individual forms[13]. Furthermore, our survey was not externally validated, and nonresponders presumably had a negative attitude toward VL and its usage in clinical practice. Despite the limitations, the current study is the first to provide data on the availability of VL, introduction into practice and patterns of use in Hungary.

Our key finding was that 65% of the responders reported availability of VL at at least one anesthesia workstation. Unfortunately, only limited data were available for comparison and were mainly from UK audit projects[11,13–15]. In 2010, Gill et al. found 57% availability of VL, while in 2017, Cook et al. described more than 90% availability of VL[11,13]. Both of the aforementioned studies examined UK hospitals. Individual responses could not be compared directly with hospital data and vice versa, but based on the aforementioned figures, the current Hungarian situation regarding the availability of VL in hospitals might be estimated to be is between the UK situations in 2010 and 2017. Hospital availability is essential for the application of VL in clinical practice. However, a well-trained anesthesiologist is the real link between available devices and patients. Therefore, from the perspective of the patient, the real availability is different and presumably lower than the hospital availability for many reasons.

The most well supplied clinical areas were surgery, the intensive care unit and traumatology, while the poorest availabilities were found in the pediatric, emergency and ear-nose-throat units, similar to a previous study[11]. In the intensive care unit, we found a lower availability rate than Cook et al. In Cook's study, they found a 54% availability rate, while we obtained a 30–44% availability rate depending on the time window[11]. Porhomayon et al. found that only 34% of the surveyed intensive care units had videolaryngoscopes contained as part of "difficult airway carts" in 2010 in the USA[16]. The lower availability of VL in pediatric units than in other units can be explained by the lower incidence of difficult intubations, fewer suitable devices and the lack of evidence of benefits[17–19]. The low availability in ear-nose-throat units might be explained by immediate access to surgical airways and the availability of fiberoptic devices. A one-gate emergency department is a new concept in Hungary, where the vast majority of patients do not need any advanced airway management; thus, airway management devices might not be the main focus there. For the sake of precise understanding of our results, we would like to highlight that by "units" and "clinical areas" we mean the subspecialties where the responding anesthesiologists works. These so called units can be located close by or far from each other regarding distance. Anesthesiologists can be permanently dedicated to these units or they can work there occasionally based on their rotation.

Eighteen devices were listed in this survey, but 44% of positive answers were related to the top three devices (KingVision, MacGrath Mac and Airtraq). In previous UK studies, the top three devices were, in order, the Airtraq, Glidescope and C-Mac[11,13]. The Airtraq occupied 50% of the market, and the aforementioned three devices accounted for 81% of overall videolaryngoscope availability in 2017 in the UK[11]. The following scopes were not reported to be available, nor were they used by the responders in clinical settings according to our results: the

Coopdech, Shikani, Upsherscope and Wuscope. These results are in accordance with the results of Cook's study[11]. Interestingly, the KingVision was found to be the leading videolaryngoscope in Hungary, although this device is almost never used by UK anesthesiologists [11,13]. Regardless of the increasing number of available videolaryngoscopes, the majority of the scopes are rarely used. Our results show that videolaryngoscope selection is mainly based on short clinical trials, the decision of the departmental lead or the price of the scope. These results are also in accordance with the results of Cook's study[11].

There is still an ongoing debate regarding the exact role of VL in airway management[2,20–22]. According to recent studies VL is preferred and successfully used to rescue failed direct laryngoscopy especially by well trained and experienced operators[2,3]. Although it is proven that VL might fail as well and it can't be considered as an ultimate solution, especially since VL can't provide oxygenation to apneic patients[23]. Furthermore, it needs to be emphasized that beyond availability of any device, the overall strategy and training of the operators seems to be far more important in airway management regarding positive outcome[24]. However, the overall attitude of our responders was positive toward the use of VL. The vast majority of the responders considered VL beneficial (98%), and 11% of them chose to use VL even for "routine" airway management. However, they generally found VL to be useful only under "special circumstances", mainly in difficult airway management scenarios, besides fiberoscopy, which was considered to be a main alternative.

According to a recent Cochrane review, the advantages of VL are limited to situations where VL is available and the user is appropriately trained and competent[2]. Appropriate training on VL should cover theoretical and practical aspects as well. Physicians need to be aware of VL technic and its role in airway management to use it in practice first on mannequins and thereafter in clinical settings. In a 2011 North American survey of residency training, VL was taught in 80% of programs and widely reported to be beneficial in teaching airway management[25,26]. Only 28% of the responding anesthesiologists used VL for teaching purposes, but the majority of users (67%) received at least some type of training regarding VL, mainly on voluntary basis and involving dolls.

## Conclusions

Based on this survey, approximately two-thirds of the Hungarian anesthesiologists have immediate access to VL, mainly in surgery, intensive care and traumatology units. The overall attitude is very positive toward VL. However, the vast majority of users prefer to use VL only in cases of difficult airway management. Even though many devices are available on the market and are known by Hungarian anesthesiologists, three to five devices are most commonly used. A particular videolaryngoscope was mainly chosen by the following methods: a short clinical trial, a decision of the departmental lead and price. A significant number of anesthesiologists reported using VL without compulsory training or any training, which needs to be improved in the future.

## Supporting information

**S1 Appendix. Questionnaire in English.**
(DOCX)

**S2 Appendix. Questionnaire in Hungarian.**
(DOCX)

**S1 Table. Primary dataset.**
(XLSX)

## Acknowledgments

The author wishes to thank the Hungarian Society of Anesthesiology and Intensive Therapy for generous support regarding the distribution of the survey. The author is also grateful for all the responses and help of Hungarian anesthesiologists, especially from the Department of Anesthesiology and Intensive Therapy, Medical School, University of Pécs, Hungary.

## Author Contributions

**Conceptualization:** Bálint Nagy, Szilárd Rendeki.

**Formal analysis:** Bálint Nagy, Szilárd Rendeki.

**Investigation:** Bálint Nagy.

**Methodology:** Bálint Nagy.

**Project administration:** Bálint Nagy.

**Supervision:** Bálint Nagy.

**Visualization:** Bálint Nagy.

**Writing – original draft:** Bálint Nagy, Szilárd Rendeki.

**Writing – review & editing:** Bálint Nagy, Szilárd Rendeki.

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
