## [Decision Letter · Decision Letter 0]

7 Aug 2019

PONE-D-19-18894

A national survey of videolaryngoscopy in Hungary

PLOS ONE

Dear Dr. Nagy,

Thank you for submitting your manuscript to PLOS ONE. After careful consideration, we feel that it has merit but does not fully meet PLOS ONE’s publication criteria as it currently stands. Therefore, we invite you to submit a revised version of the manuscript that addresses the points raised during the review process.

We would appreciate receiving your revised manuscript by Sep 21 2019 11:59PM. To enhance the reproducibility of your results, we recommend that if applicable you deposit your laboratory protocols in protocols.io, where a protocol can be assigned its own identifier (DOI) such that it can be cited independently in the future. For instructions see: http://journals.plos.org/plosone/s/submission-guidelines#loc-laboratory-protocols

We look forward to receiving your revised manuscript.

Kind regards,

Mohamed R El-Tahan, MD

Academic Editor

PLOS ONE

Journal Requirements:

2. Please include copies of the survey questions or questionnaires used in the study, in both the original language as well as English, as Supporting Information, or include a citation if they have been published previously.

4. Please provide additional details regarding participant consent. In the ethics statement in the Methods and online submission information, please ensure that you have specified (1) whether consent was suitably informed and (2) what type you obtained (for instance, written or through a question on the questionnaire). If the need for explicit consent was waived by the ethics committee, please include this information.

Reviewers' comments:

Reviewer's Responses to Questions

**Comments to the Author**

1. Is the manuscript technically sound, and do the data support the conclusions?

Reviewer #1: Yes

Reviewer #2: Partly

2. Has the statistical analysis been performed appropriately and rigorously? 

Reviewer #1: Yes

Reviewer #2: Yes

3. Have the authors made all data underlying the findings in their manuscript fully available?

Reviewer #1: Yes

Reviewer #2: Yes

4. Is the manuscript presented in an intelligible fashion and written in standard English?

Reviewer #1: Yes

Reviewer #2: Yes

5. Review Comments to the Author

Reviewer #1: Paper by Nagy and Rendeki covers an interesting topic in era of large debate and discussion in era of videolaryngoscopes.

English is fluent, methodology correct, though some more data would have been interesting for evaluation.

Discussion is adequate, and some more (minor) issues could be addressed (see below).

References are adequate and updated; consider to add some more as per comments below.

Appendix not available for consultation.

Page 4, line 9, consider adding a more comprehensive review as reference: Frova G, SORBELLO M. Algorithms for difficult airway management: a review. Minerva Anestesiologica 2009, 75(4): 201-209.

Page 5 line 2: please state precisely date of initiation of survey.

Page 5, lines 5-7: the sentence “The study presumed that the connection between the patient and the device used for airway management is the anesthesiologist.” is clear, but I would reformulate; I would also add a note whether the VLS were used by non-anesthesiologists. In fact, as from abstract, only 805 of responders were specialists and only 68% were anesthesiologists. Please also address in text if in your national educational program, anesthesia and intensive care are different specialties or different training programs.

Page 5: The authors list both videolaryngoscopes and optical/digital stylets (Bonfils, Levithan, Shikani), modified classic laryngoscopes (Upsherscope, Bullard, WuScope) and other devices (C-trach). I would address this information in title and rest of text, probably changing into “videolaryngoscopes and alternative intubation devices”. Also, presentation of data (either tables/figures) could take account of this classification, for the sake of precision.

Page 9 line 5: any data about failure rate for VLS? Or some information about decision to use fiberoptic scope rather than VLS?

Page 10, line 13: provide reference for UK NAP4.

Some points I would furtherly address in discussion:

- Which role for bougies as adjunct in airway management? Any data?

- I would underline also with references that VLS might fail (Aziz MF, Brambrink AM, Healy DW, Willett AW, Shanks A, Tremper T, Jameson L, Ragheb J, Biggs DA, Paganelli WC, Rao J, Epps JL, Colquhoun DA, Bakke P, Kheterpal S. Success of Intubation Rescue Techniques after Failed Direct Laryngoscopy in Adults: A Retrospective Comparative Analysis from the Multicenter Perioperative Outcomes Group. Anesthesiology. 2016; 125: 656-66.), that they cannot be a definitive solution, as they do not provide oxygenation in apneic patient (Sgalambro F, Sorbello M. Videolaryngoscopy and the search for the Holy Grail. Br J Anaesth. 2017 Mar 1;118(3):471-472.) and finally that strategy needs to be considered a first important point, independently on availability of whichever device (Sorbello M, Afshari A, De Hert S. Device or target? A paradigm shift in airway management with implications for guidelines, clinical practice and teaching. European Journal of Anaesthesiology, 2018 Nov; 35 (11): 811-814.)

- Some further comments about importance of training, also considering that success rate with VLS is related to experience in their daily use.

Reviewer #2: Dear Authors,

I read with interest your manuscript. It is an interesting study, nevertheless I have some concerns .

General comments

As mentioned by the authors, this is the first study on VLs in the so-called “Eastern-European” countries (I would eliminate this denomination, leaving just Hungary, not introducing a further biasing term, we don’t know how is the situation in other neighboring countries),

There are several confusing points that should be clarified:

- The “units” that the authors mention – surgery, traumatology, intensive care, ENT, …these are probably the subspecialities where the responding anesthesiologists are working, inside the OR or outside it (ICU), isn’t it? That needs more precision in the text, for correct understanding by anesthesiologists who are not necessarily familiar with the Hungarian system.

- There is a very huge availability of VLs on the market generally, and in Hungary too, depending of course on multiple factors. I would skip the repetition several times of very rarely or never used devices, and leave just the most used ones.

- Videolaryngoscopy means better visualization, not necessarily better intubation – in what consists exactly the training for VL use that the authors mention several times? Knowing that VLs exist and how they work, or using them in mannequins and thereafter in clinical settings?

- If I understand correctly, the exact purpose of the study was to evaluate the proportion of anesthesiologist using VLs, the most used VLs, and the time needed to have them ready. That should be stated more clearly.

- Moreover, what this study will teach to other anesthesiologists, either from Hungary or from abroad? What is the clear message emanating from this study.

Specific comments

Abstract

Results

This phrase is not very clear, please reformulate it. It’s not clear that they were either anesthesiologists participating in the study, or surgeons or trauma specialists?

Were there 324 or 210 responders actually? In the abstract is not clear (even if in the final text of the manuscript is OK)

“”…hundred and ten (65%) responders had access to various videolaryngoscopes and

were mainly from surgery, intensive care and traumatology units. No responders

reported the availability of eight videolaryngoscopes out of the eighteen listed devices,

and 32% of the responders had never used any videolaryngoscope in clinical settings.”

Manuscript

Results

In the table, including the “professional activities” how people who are not doing clinical anesthesia (education, administration, other) could use VLs?

In the same table, it would be interesting to define the activities of the several types of hospitals cited, in order to have an idea which type of surgery/ in which cases the VLs are used – obviously, difficult intubation occurs more frequently in the delivery room for CS, in facial trauma patients, in ENT surgery, ….

Availability of VLs

If there were several brands of VLs that were never reported to have been used, they should not be included in the study.

Patterns of use

“One hundred and four responders (32%) said that they had never ever used any

videolaryngoscope in clinical settings. A similar number of colleagues (n=118, 36%) stated that they never use VL. Only 39% (n=126) confirmed that they used VL at least once per month.” – I don’t understand this phrase –“never used in clinical settings vs they never use VL” – what’s the difference?

Discussion

The discussion is quite clear, and its understanding is good, nevertheless, I would mention the total number of anesthesiologists in the materials and methods section, as well as the proportion of responders in the results section, even if it’s repeated in the discussion again.

6. PLOS authors have the option to publish the peer review history of their article (what does this mean?). If published, this will include your full peer review and any attached files.

Reviewer #1: Yes: Massimiliano Sorbello

Reviewer #2: No

---

## [Author Response · Author response to Decision Letter 0]

26 Aug 2019

Prof. Dr. Mohamed R El-Tahan 

Academic Editor

PLOS ONE

August 26th, 2019 

PONE-D-19-18894

Dear Prof. Dr. Mohamed R El-Tahan,

First of all, we would like to say thank you for handling our manuscript as an academic editor of PLOS One! Based on the received valuable recommendations and comments, we made significant efforts to revise our manuscript. Please find our detailed answers for you and for the reviewers below. 

Journal Requirements:

We made corrections during revision to fulfill all the above mentioned criteria regarding manuscript style. Please find the revised version of our manuscript for all the corrections we made.

2. Please include copies of the survey questions or questionnaires used in the study, in both the original language as well as English, as Supporting Information, or include a citation if they have been published previously.

We included the questionnaires as Supporting Information in English and in Hungarian as well (S1 and S2 Appendixes). The following sentence of the „Materials and Methods” refers to the questionnaires in the revised manuscript: “Questions relevant to the availability, use and introduction of VL are shown respectively in English and in Hungarian as supporting information in S1 and S2 Appendixes”

We included captions for Supporting Information files according to the recommended guideline.

4. Please provide additional details regarding participant consent. In the ethics statement in the Methods and online submission information, please ensure that you have specified (1) whether consent was suitably informed and (2) what type you obtained (for instance, written or through a question on the questionnaire). If the need for explicit consent was waived by the ethics committee, please include this information.

We included additional details in the revised version regarding consent. The following new sentence of the „Materials and Methods” refers to the consent of the participants: „Informed consent regarding participation and publishing was obtained from the participants through a question of the questionnaire.”

Since there are no legal or ethical restrictions on sharing of the data set of our study, we made it freely available in the revised version of our manuscript as Supporting Information. The following new sentence of the „Results” refers to the data set: „In total, 324 completed forms were returned without duplicates (S3 Table)”

Dear Prof. Dr. Massimiliano Sorbello,

Thank you for your valuable time you spent on providing precise and professional review regarding our manuscript. We are really grateful for all of your valuable recommendations and comments, and we made significant efforts to correct the manuscript accordingly! Please find our detailed answers below!

Paper by Nagy and Rendeki covers an interesting topic in era of large debate and discussion in era of videolaryngoscopes. English is fluent, methodology correct, though some more data would have been interesting for evaluation. Discussion is adequate, and some more (minor) issues could be addressed (see below). References are adequate and updated; consider to add some more as per comments below.

Appendix not available for consultation.

We included appendixes as supporting information to the submission of the revised manuscript regarding the questionnaire. The following sentence of the „Materials and Methods” refers to the questionnaires in the revised manuscript: “Questions relevant to the availability, use and introduction of VL are shown consecutively in English and in Hungarian as supporting information in S1 and S2 Appendixes”

Page 4, line 9, consider adding a more comprehensive review as reference: Frova G, SORBELLO M. Algorithms for difficult airway management: a review. Minerva Anestesiologica 2009, 75(4): 201-209.

Thank you for the recommendation! We added the recommended article as a new reference!

Page 5 line 2: please state precisely date of initiation of survey.

Thank you! For the sake of precision we included the date of initiation of the survey as you recommended. We modified the abstract and the „materials and methods” as well. The new sentence: „The survey was conducted between 01.01.2018 and 31.12.2018.”

Page 5, lines 5-7: the sentence “The study presumed that the connection between the patient and the device used for airway management is the anesthesiologist.” is clear, but I would reformulate; I would also add a note whether the VLS were used by non-anesthesiologists. In fact, as from abstract, only 805 of responders were specialists and only 68% were anesthesiologists. Please also address in text if in your national educational program, anesthesia and intensive care are different specialties or different training programs.

Thank you! Indeed, all the points you mentioned above might be misleading without further clarification. In abstract, we originally intended to say that 80% of the responders were specialists (20% were trainees), while 68% were involved rather in anesthesia practice than other listed professional activities (intensive care, education, etc.). We modified the abstract to clarify: „Responders were mainly males (58%), specialists (80%) and those involved mainly in anesthesia practice (68%)…”. Furthermore, we added a few sentences to „Materials and Methods” regarding the anesthesia educational program and VL usage by non-anesthesiologists: „Anesthesiology and intensive therapy is a combined, five years long training program in Hungary, thus all the anesthesiologists are intensive care physicians as well. In this study, we collected answers from anesthesiologists and anesthesiology trainees only, even though we aware of the fact, that other physicians like emergency and internal medicine doctors might occasionally use VL for advanced airway management. Although, still anesthesiologists are responsible for advanced airway management is Hungary in the vast majority of cases.” 

Page 5: The authors list both videolaryngoscopes and optical/digital stylets (Bonfils, Levithan, Shikani), modified classic laryngoscopes (Upsherscope, Bullard, WuScope) and other devices (C-trach). I would address this information in title and rest of text, probably changing into “videolaryngoscopes and alternative intubation devices”. Also, presentation of data (either tables/figures) could take account of this classification, for the sake of precision.

Thank you for highlighting an important point regarding classification! We approve that not all the listed devices are classic videolaryngoscopes, thus we modified the title of the manuscript as you recommended. Furthermore, we added a few lines to the „Materials and Methods” to highlight this issue for the readers as well: „We would like to emphasize here that not all of the aforementioned devices are classic videolaryngoscopes. Bonfils, Levithan and Shikani are optical/digital stylets, Upsherscope, Bullard and WuScope are modified classic laryngoscopes, while C-trach is also a different kind of intubation device. However, to avoid confusion, we prefer to refer these devices also as VL’s throughout this study similarly to a recent major evaluation of Cook and Kelly…”. However, we considered to use the study of Cook and Kelly as a base for our evaluation, which also referred the listed devices as videolaryngoscopes. (Cook TM, Kelly FE. A national survey of videolaryngoscopy in the United Kingdom. Br J Anaesth. 2017;118: 593–600.) 

Page 9 line 5: any data about failure rate for VLS? Or some information about decision to use fiberoptic scope rather than VLS?

Unfortunately, we collected no data in this study on the VL failure rate and decision to use a fiberoptic scope over VL. As far as we know, there is an ongoing national audit project on airway management in general in Hungary, which might be able to answer these important and interesting questions. 

Page 10, line 13: provide reference for UK NAP4.

Thank you for the recommendation! References regarding UK NAP4 are added.

Some points I would furtherly address in discussion:

- Which role for bougies as adjunct in airway management? Any data?

Since we collected no data on airway adjuncts and we generally aimed to explore national data on the availability of VL, introduction into practice and patterns of use in Hungary, we would be a bit concerned to discuss airway management adjuncts in this manuscript. Although it would be important and interesting to know more about this topic in details at national level. Hopefully, the above mentioned ongoing national audit project will provide data on this topic soon. 

- I would underline also with references that VLS might fail (Aziz MF, Brambrink AM, Healy DW, Willett AW, Shanks A, Tremper T, Jameson L, Ragheb J, Biggs DA, Paganelli WC, Rao J, Epps JL, Colquhoun DA, Bakke P, Kheterpal S. Success of Intubation Rescue Techniques after Failed Direct Laryngoscopy in Adults: A Retrospective Comparative Analysis from the Multicenter Perioperative Outcomes Group. Anesthesiology. 2016; 125: 656-66.), that they cannot be a definitive solution, as they do not provide oxygenation in apneic patient (Sgalambro F, Sorbello M. Videolaryngoscopy and the search for the Holy Grail. Br J Anaesth. 2017 Mar 1;118(3):471-472.) and finally that strategy needs to be considered a first important point, independently on availability of whichever device (Sorbello M, Afshari A, De Hert S. Device or target? A paradigm shift in airway management with implications for guidelines, clinical practice and teaching. European Journal of Anaesthesiology, 2018 Nov; 35 (11): 811-814.)

- Some further comments about importance of training, also considering that success rate with VLS is related to experience in their daily use.

Thank you for recommending to emphasize these important points! We added details to the “Discussion” with references as you recommended: “According to recent studies VL is preferred and successfully used to rescue failed direct laryngoscopy especially by well trained and experienced operators[2,3]. Although it is proven that VL might fail as well and it can’t be considered as an ultimate solution, especially since VL can’t provide oxygenation to apneic patients[23]. Furthermore, it needs to be emphasized that beyond availability of any device, the overall strategy and training of the operators seems to be far more important in airway management regarding positive outcome[24].”.

Dear Reviewer #2,

Thank you for your time and for all the valuable comments and recommendations! We made significant efforts to revise our manuscript accordingly. Please find our detailed answers below!

I read with interest your manuscript. It is an interesting study, nevertheless I have some concerns .

General comments

As mentioned by the authors, this is the first study on VLs in the so-called “Eastern-European” countries (I would eliminate this denomination, leaving just Hungary, not introducing a further biasing term, we don’t know how is the situation in other neighboring countries),

Thank you! We eliminated the term “Eastern-European” from the revised version of our manuscript to prevent further bias.

There are several confusing points that should be clarified:

- The “units” that the authors mention – surgery, traumatology, intensive care, ENT, …these are probably the subspecialities where the responding anesthesiologists are working, inside the OR or outside it (ICU), isn’t it? That needs more precision in the text, for correct understanding by anesthesiologists who are not necessarily familiar with the Hungarian system.

Thank you! We added the following lines to the „Discussion” to clarify this point: „For the sake of precise understanding of our results, we would like to highlight that by “units” and “clinical areas” we mean the subspecialties where the responding anesthesiologists works. These so called units can be located close by or far from each other regarding distance. Anesthesiologists can be permanently dedicated to these units or they can work there occasionally based on their rotation.”

- There is a very huge availability of VLs on the market generally, and in Hungary too, depending of course on multiple factors. I would skip the repetition several times of very rarely or never used devices, and leave just the most used ones.

Thank you for this recommendation! Since it is the first report on VL from Hungary, we prefer to show not only the popular VLs, but also the rarely/never used ones. Guidelines on VL position in airway management are quite clear. However, data on VL selection are sparsely published. We consider useful and interesting to show, that a device like KingVision, which is almost never used for example in the UK, is popular in Hungary, while few other devices like Coopdech, Shikani, Upsherscope and Wuscope are equally neglected in the UK and in Hungary as well. Even though there is no clear evidence in general on choosing one VL over another. We would appreciate if you let us to present our results on rarely/never used VLs as well. 

- Videolaryngoscopy means better visualization, not necessarily better intubation – in what consists exactly the training for VL use that the authors mention several times? Knowing that VLs exist and how they work, or using them in mannequins and thereafter in clinical settings?

Thank you for raising this important point! We consider both of the aforementioned parts of the training are equally important. We included the following sentences to the „Discussion” for clarification: „Appropriate training on VL should cover theoretical and practical aspects as well. Physicians need to be aware of VL technic and its role in airway management to use it in practice first on mannequins and thereafter in clinical settings.”

- If I understand correctly, the exact purpose of the study was to evaluate the proportion of anesthesiologist using VLs, the most used VLs, and the time needed to have them ready. That should be stated more clearly.

Thank you! We tried to clarify the exact purpose of our study with the following sentence of the introduction: „Since data on the availability of VL are rarely published, our primary objective was to explore national data on the availability of VL, introduction into practice and patterns of use in Hungary to gain data on the proportion of anesthesiologist using VL, the most used VL, and the time needed to have VL readily available in clinical settings.”

- Moreover, what this study will teach to other anesthesiologists, either from Hungary or from abroad? What is the clear message emanating from this study.

In „Conclusions” section we aimed to highlight the key findings of our study like attitude is positive, the majority of devices is known, only a few type of VLs are used in clinical settings (mainly for difficult airway management) and availability is significant. As a message of our study, we recommended that availability and training need to be improved further. We would appreciate your help to emphasize the message of our study If you are concerned that it is not clear for the readers.

Specific comments

Abstract

Results

This phrase is not very clear, please reformulate it. It’s not clear that they were either anesthesiologists participating in the study, or surgeons or trauma specialists?

Thank you for highlighting this important point! We aimed to clarify this in the revised manuscript with the following new sentence of the „Materials and Methods”: „In this study, we collected answers from anesthesiologists and anesthesiology trainees only, even though we aware of the fact, that other physicians like emergency and internal medicine doctors might occasionally use VL for advanced airway management.”

Were there 324 or 210 responders actually? In the abstract is not clear (even if in the final text of the manuscript is OK)

“”…hundred and ten (65%) responders had access to various videolaryngoscopes and

were mainly from surgery, intensive care and traumatology units. No responders

reported the availability of eight videolaryngoscopes out of the eighteen listed devices,

and 32% of the responders had never used any videolaryngoscope in clinical settings.”

Thank you! The abstract might be misleading to the reader, thus we modified the „Results” section to make the number of responders clearer: „In total, 324 duly completed forms were returned and analyzed.” 

Manuscript

Results

In the table, including the “professional activities” how people who are not doing clinical anesthesia (education, administration, other) could use VLs?

Thank you! Indeed, this might look ambivalent or even impossible, but the question regarding professional activities in the questionnaire was the following: „Which one of the followings is the most specific to your daily professional activity? (Single best answer!)”. In total, 4/324 anesthesiologist answered that education or other (non-patient-related) activity is the most specific to his/her daily professional activity. These anesthesiologists still practice their jobs, but they are mainly involved rather in something else. 

In the same table, it would be interesting to define the activities of the several types of hospitals cited, in order to have an idea which type of surgery/ in which cases the VLs are used – obviously, difficult intubation occurs more frequently in the delivery room for CS, in facial trauma patients, in ENT surgery, ….

Thank you for raising this interesting point! Unfortunately, we didn’t collect any data from responders regarding the activities of their hospitals and since our survey collected completely anonymous answers, we don’t have the name of the hospitals we received answers from. In Hungary, we have no national standards regarding the capabilities of each hospital type. For example, many county hospitals have neurosurgical, maxillofacial surgical, etc. capabilities, but not all. So, based on the data we have, we are unable to provide details beyond hospital types. 

Availability of VLs

If there were several brands of VLs that were never reported to have been used, they should not be included in the study.

Thank you! Retrospectively, we completely agree with this point of you, that VLs never reported to have been used, should be omitted from our study. Although, we considered to use the study of Cook and Kelly as a base for our evaluation, which also used almost the same list of devices (Cook TM, Kelly FE. A national survey of videolaryngoscopy in the United Kingdom. Br J Anaesth. 2017;118: 593–600.). Furthermore, we didn’t expect exactly the same results as previous studies showed earlier. This presumption is at least partially proven: KingVision found to be popular in Hungary, while in the United Kingdom it is almost never used. However, we will strongly consider this recommendation for further similar studies. 

Patterns of use

“One hundred and four responders (32%) said that they had never ever used any videolaryngoscope in clinical settings. A similar number of colleagues (n=118, 36%) stated that they never use VL. Only 39% (n=126) confirmed that they used VL at least once per month.” – I don’t understand this phrase –“never used in clinical settings vs they never use VL” – what’s the difference?

We agree that the above mentioned phrases might be confusing to the reader. We intended to say that 32% „never ever used” VL in his/her career in clinical settings at all, while 36% don’t use VL in general. So there is a 4% difference, which might mean that even though they tried VL at least once, they didn’t find it useful. However, we approve that this all can be a bit confusing, thus we omitted the following sentence from the revised version of our manuscript: „A similar number of colleagues (n=118, 36%) stated that they never use VL.” 

Discussion

The discussion is quite clear, and its understanding is good, nevertheless, I would mention the total number of anesthesiologists in the materials and methods section, as well as the proportion of responders in the results section, even if it’s repeated in the discussion again.

Thank you for this recommendation! We added the following lines to the „Materials and Methods” and to the „Results” sections respectively: „We aimed to reach all the 1567 anesthesiologists of Hungary.” and „Response rate was 21%.” 

We hope that after the significant work we have done to fulfill all the requests, you will find our manuscript suitable for publication in PLOS ONE!

Yours sincerely,

Bálint Nagy, M.D. PhD

Department of Anaesthesiology and Intensive Therapy

University of Pécs, Hungary

HU-7624 Pécs, Ifjúság str. 13, Hungary 

Tel: + 36 72 536 000, Fax: + 36 72 533 117

E-mail: balintjanosnagy@yahoo.com

---

## [Decision Letter · Decision Letter 1]

26 Sep 2019

A national survey of videolaryngoscopes and alternative intubation devices in Hungary

PONE-D-19-18894R1

Dear Dr. Nagy,

We are pleased to inform you that your manuscript has been judged scientifically suitable for publication and will be formally accepted for publication once it complies with all outstanding technical requirements.

With kind regards,

Mohamed R. El-Tahan, MD

Academic Editor

PLOS ONE

Additional Editor Comments (optional):

Reviewers' comments:

Reviewer's Responses to Questions

**Comments to the Author**

1. If the authors have adequately addressed your comments raised in a previous round of review and you feel that this manuscript is now acceptable for publication, you may indicate that here to bypass the “Comments to the Author” section, enter your conflict of interest statement in the “Confidential to Editor” section, and submit your "Accept" recommendation.

Reviewer #1: All comments have been addressed

Reviewer #2: All comments have been addressed

2. Is the manuscript technically sound, and do the data support the conclusions?

Reviewer #1: Yes

Reviewer #2: Yes

3. Has the statistical analysis been performed appropriately and rigorously? 

Reviewer #1: Yes

Reviewer #2: Yes

4. Have the authors made all data underlying the findings in their manuscript fully available?

Reviewer #1: Yes

Reviewer #2: Yes

5. Is the manuscript presented in an intelligible fashion and written in standard English?

Reviewer #1: Yes

Reviewer #2: Yes

6. Review Comments to the Author

Reviewer #1: I am happy with all changes, thank to the Authors for their efforts.

Please just check a final round for typos and formats

Reviewer #2: I would like to thank the Authors of this manuscript for their answers and corrections. They have done a great job!

7. PLOS authors have the option to publish the peer review history of their article (what does this mean?). If published, this will include your full peer review and any attached files.

Reviewer #1: No

Reviewer #2: Yes: Laszlo L. SZEGEDI, M.D., PhD, Brussels, Belgium

---

## [Editor Report · Acceptance letter]

2 Oct 2019

PONE-D-19-18894R1 

A national survey of videolaryngoscopes and alternative intubation devices in Hungary 

Dear Dr. Nagy:

I am pleased to inform you that your manuscript has been deemed suitable for publication in PLOS ONE. Congratulations! Your manuscript is now with our production department. 

With kind regards,

on behalf of

Professor Mohamed R. El-Tahan 

Academic Editor

PLOS ONE